# Human Papillomavirus and Male Infertility: What Do We Know?

**DOI:** 10.3390/ijms242417562

**Published:** 2023-12-16

**Authors:** Arianna Sucato, Michela Buttà, Liana Bosco, Leonardo Di Gregorio, Antonio Perino, Giuseppina Capra

**Affiliations:** 1Department of Health Promotion, Mother and Child Care, Internal Medicine and Medical Specialties (ProMISE) “G. D’Alessandro”, University of Palermo, Piazza delle Cliniche 2, 90127 Palermo, Italy; sucato.arianna@gmail.com (A.S.); michela.butta@unipa.it (M.B.); 2Section of Biology and Genetics, Department of Biomedicine, Neuroscience and Advanced Diagnostics (Bi.N.D), University of Palermo, 90133 Palermo, Italy; liana.bosco@unipa.it; 3Unit of Urology, Polyclinic Hospital, Via del Vespro 133, 90127 Palermo, Italy; leondigregorio@gmail.com; 4IVF Unit, Department of Obstetrics and Gynecology, Villa Sofia Cervello Hospital, University of Palermo, 90146 Palermo, Italy; antonio.perino@unipa.it; 5UOC of Microbiology and Virology, Polyclinic Hospital, Via del Vespro 133, 90127 Palermo, Italy

**Keywords:** infertility, HPV infection, sperm parameters, ART, vaccination, HPV, male infertility

## Abstract

In recent years, increasing attention has been paid to understanding the causes of infertility, which is being recognized as a growing health problem affecting large numbers of couples worldwide. Male infertility is a contributing factor in approximately 30–40% of cases, and one of its etiological causes is sexually transmitted infections (STIs). Among sexually transmitted pathogens, human papillomavirus (HPV) can contribute in various ways to the failure of spontaneous and assisted reproduction, acting in the different phases of conception, especially in the early ones. In particular, HPV infection can affect sperm DNA integrity, sperm motility, count, viability, and morphology and can induce the production of anti-sperm antibodies (ASAs). In this narrative review, we aimed to provide an overview of existing research on the potential adverse effects of HPV infection on male reproductive health. Furthermore, we analyzed how limiting the spread of the infection, particularly with gender-neutral vaccination, could be a possible therapeutic tool to counteract male and female fertility problems.

## 1. Introduction

Infertility, which affects about 10–15% of couples worldwide, is a major public health concern, and male factors contribute substantially in approximately 30–40% of cases [1,2]. Male infertility is a multifactorial condition, and among the causes implicated with this status, it is possible to find genetic disorders, anatomical defects, systemic diseases, sexually transmitted infections, varicocele, oxidative stress, and erectile dysfunction as well as lifestyle factors, including smoking, diet, and radiation and xenobiotic exposure [1,3,4]. All these variables can negatively influence the processes involved in adequate sperm production, such as spermatogenesis, epididymal maturation, sperm storage, sperm transport, and accessory gland function [5]. Nevertheless, about 50% of infertile men are subject to idiopathic infertility, in which only oligospermia, asthenospermia, teratozoospermia, or other alterations are found in sperm [3]. 

There is considerable evidence that certain sexually transmitted infections (STI), including those caused by human immunodeficiency virus (HIV), Chlamydia trachomatis (CT), and human papillomavirus (HPV), can affect fertility [6].

HPV infection is the most common sexually transmitted infection, and almost all sexually active men and women are at risk of acquiring it during their lifetime [7,8]. The most common way in which these infections are spread is by direct contact of the skin or mucous membranes, particularly through vaginal or anal intercourse. In addition, non-penetrative sexual activities such as oral–genital, genital–genital, and hands-to-genital are potential routes of transmission [9,10]. 

More than 200 different types of HPV have been identified, including those acknowledged by the International Agency for Research on Cancer (IARC) as high-risk (HPV16, 18, 31, 33, 35, 39, 45, 51, 52, 56, 58, 59, 68), associated with cancers of the cervix, vagina, vulva, anus, head and neck, and penis; and those recognized as low-risk (HPV6 and 11) involved in the development of benign lesions, such as genital warts or condyloma acuminata [3,11,12]. 

Most of the time, the infection is completely asymptomatic and is cleared by the immune system within 12–24 months without any clinical sequelae [8]. When this does not occur, a persistent infection is settled, which is thought to be a prerequisite for the development of potential neoplastic hyperproliferative lesions [8]. 

In men, it has been discovered that HPV is present in semen as well as in several genital areas, including the penile shaft, glans, corona, scrotum, perianal, and anal regions [13,14]. HPV can affect every cellular component of the seminal fluid and could impair sperm parameters, including sperm count, motility, genomic integrity, morphology and concentration, leading to male infertility [10]. Notwithstanding, it is not known what the exact mechanism of HPV infection in spermatozoa is nor what role the infected cells play in the transmission of the virus [15]. For instance, recent studies have shown that HPV virions are able to bind to the equatorial segment of the sperm head through the interaction between L1 viral capsid protein and the proteoglycan syndecan-1 [16,17,18].

In addition, it has been speculated that infected sperm can transmit the infection to the cervix via fecundation, and as an ascending infection can even reach the placenta, causing miscarriages and infertility in couples [19,20]. Another way in which HPV could affect sperm quality is through the induction of the production of anti-sperm antibodies (ASAs), which would be responsible for sperm motility impairment [21].

The development of vaccines has been a major step forward in efficiently counteracting HPV infection [10]. Indeed, prophylactic HPV vaccination was proposed as a strategy for improving male fertility because of the ability of vaccine immunization to reduce semen infection [22]. 

In the following sections, we will focus on the influence of HPV infection on male fertility, describing its ability to affect sperm parameters and sperm DNA integrity. We will also explore the theory of HPV-mediated induction of anti-sperm antibodies (ASAs) as well as the possible differences between high-risk HPV (hrHPV) and low-risk HPV (lrHPV) seminal infections. Finally, we will analyze the implications of the infection for assisted reproductive technology (ART), the potential impact of vaccination on counteracting male infertility, and the possible involvement of the positivity of both partners in infertility problems. 

Given the intricacy of the different findings in the literature about the relationship between HPV and male infertility, we have attempted to review the literature by providing a summary perspective of the various existing points of view. To achieve this, we used PubMed with the keywords HPV, sperm parameters, ART, sperm DNA fragmentation, ASA, HPV vaccination, and male infertility. The search included all the studies conducted to date, mainly between 1997 and 2023, except two studies from 1954 and 1959, which aimed to attribute paternity to the discovery of the role of ASA.

The novel aspect of this narrative review is that we aim to provide up-to-date insight into the quality of evidence regarding the involvement of HPV infection in male infertility and to suggest ways to help manage infections and potential fertility problems.

## 2. HPV Impact on Sperm Parameters

The World Health Organization 2021 (WHO) manual defines the evaluation of sperm parameters, such as sperm count, viability, motility, morphology, and seminal fluid composition as the principal approach for semen quality assessment, as these are crucial to male fertility [23,24]. HPV infection has been linked in several studies to a decrease in progressive sperm motility and changes in immotile sperm rate, sperm morphology and concentration, although these correlations remain controversial [8,14,25,26]. The first work to report HPV-related alterations in sperm motility was conducted in 1997 by Lai et al. [27], who observed a lower curvilinear and straight-line velocity and a reduced mean amplitude of lateral head displacement of hrHPV-infected sperm cells (Table 1).

In the following years, some studies confirmed these observations [18,32], whilst others [34,35] rejected these hypotheses, arguing that there is no association between HPV semen infection and impaired semen function. The detection of HPV-DNA in semen would therefore only be a secondary effect of the desquamation of penile-HPV-infected keratinocytes [36]. Damke et al. [24] conducted a study which demonstrated a correlation between HPV semen infection, mostly with multiple HPV types, and the detection of hypospermia, abnormal viscosity, and increased seminal pH (Table 1). In contrast, several works [25,34] found no clear relationship between HPV infection and alteration in seminal volume or concentration. Such disagreement could be related to the different sample sizes or methodologies used in the various studies. For instance, Damke et al. [24] based their study on 229 semen samples, whereas Rintala et al. [34] based theirs on 65 semen samples. From a methodological point of view, both studies based the semen analysis on the WHO criteria, but they used different genotyping techniques. Rintala et al. [34] used hybridization with digoxigenin-labeled high-risk HPV oligoprobes, identifying 12 genotypes; Damke et al. [24] used PCR-RFLP, which allowed the identification of a higher number of genotypes (39 genotypes). The lower number of samples and genotypes analyzed by Rintala et al. [34] compared to Damke et al. [24] may explain the lack of correlation found by the latter. 

In terms of semen composition, semen generally has a pH of around 7.72 and must contain appropriate concentrations of citrate, potassium, sodium, calcium, magnesium, zinc, glucose, fructose, albumin, and proteins [37]. Since the volume of seminal vesicles and prostate secretions affects the concentration of sperm in the ejaculate, it also affects fertility and pregnancy rates [23]. Particularly, seminal vesicles produce fructose, the main sugar involved in the metabolic processes and motility of sperm, which has also a fundamental role in zinc chelation, fertilization, and sperm chromatin condensation. Meanwhile, the prostate gland is bound to the production of zinc and citric acid. The latter, whose levels are regulated by testosterone, is thought to be the main ligand of zinc, a key element in the regulation of sperm motility, germ cell maintenance, and spermatogenesis progression [38]. It has been suggested that in HPV-infected males, zinc production is not enough to condense sperm, which may be correlated with male infertility [10]. Indeed, Damke et al. [24] showed that men with HPV-positive semen may have altered proportions of prostate and seminal vesicle secretions, both associated with glandular dysfunction, which would have negative effects on fertility. Moreover, the detection in a 2013 study [39] of HPV-DNA in both epithelial and non-epithelial semen cells, as well as in semen leukocytes, supported the idea that HPV does not exclusively infect sperm cells. It can be concluded that the heterogeneous results regarding the relationship between seminal HPV-positivity and sperm quality could be due to the different targets of infection [40]. 

## 3. Correlation between HPV and Sperm DNA Fragmentation

Sperm DNA fragmentation (SDF) is a form of sperm nucleic acid damage that occurs before or after ejaculation and consists of double-stranded or single-stranded breaks [41]. At present, three main mechanisms are considered responsible for sperm DNA damage, namely impaired sperm chromatin maturation, unnatural sperm cell apoptosis, and oxidative stress [42,43]. 

The delivery of an undamaged paternal genome from the spermatozoa to the egg cell is essential for the correct development of the embryo [43,44]. Thus, there is a correlation between sperm DNA damage and impaired fertilization, lower embryo quality, decreased pregnancy rates, and high rates of pregnancy loss following in vitro fertilization (IVF) [41,45,46,47,48].

In light of this evidence, for the first time, the latest edition of the WHO laboratory manual for the examination/processing of human semen has included methods for evaluating the DNA fragmentation index (DFI) as an indicator of the measure of sperm DNA integrity [23]. Based on the data available in the literature, a DFI > 30% is considered predictive of infertility, while a value below 20% is considered physiological, although the WHO guidelines do not specify cut-off values [10,49]. Several techniques are considered suitable for assessing SDF, including terminal deoxynucleotidyl transferase (dUTP) nick-end-labeling (TUNEL) assay, single cell gel electrophoresis (Comet) assay, acridine orange flow cytometry (AO FCM) assay, and sperm chromatin dispersion (SCD) test (Figure 1) [23]. 

To the best of our knowledge, the integrity of sperm DNA is constantly affected by endogenous and exogenous elements. For example, double-strand breaks are physiologically induced during spermatogenesis to facilitate meiotic crossover and during spermiogenesis to facilitate histone-protamine replacement [49,50]. Nonetheless, other factors that are not always physiological, such as lifestyle (e.g., alcohol, smoking, drugs, diet), exposure to pollutants, diseases, aging, and infections can increase levels of SDF [8,43]. Specifically, infections of the genital tract in males lead to leukocytospermia and elevate oxidative stress in semen, resulting in DNA breakage in sperm [43]. Several studies have described the disruptive effect of infection with certain pathogens on sperm DNA integrity. These include mycoplasma, Chlamydia trachomatis, hepatitis B virus, and HPV, although for the latter, the literature’s data are not in agreement with each other [32,51,52]. Connelly and colleagues [32] showed that DNA fragmentation driven by fragments of the E6–E7 genes of HPV16 or 31 caused genomic DNA breaks and increased sperm apoptosis (Table 1). This finding was subsequently confirmed by Lee et al. [33], who demonstrated exonic modification of the p53 gene by the E6–E7 DNA fragments of HPV16 and 18, and Moreno-Sepulveda and Rajmil [28], who showed an increased risk of DFI > 30% in patients with HPV seminal infection (Table 1).

Despite this, some studies in the current literature have failed to find an association between a DFI value greater than 30% and the detection of HPV DNA in semen [17,53]. Furthermore, some years later, a correlation between the expression of an isoform of the E6 protein and the enhancement of oxidative stress-induced cellular DNA damage was highlighted [54]. In this context, Kato et al. [55] reported higher levels of superoxide dismutase (SOD), an antioxidant enzyme, in the seminal plasma of HPV-positive men compared to HPV-negative ones, suggesting the possible involvement of reactive oxygen species (ROS) in sperm DNA damage. ROS are physiologically produced in the cell microenvironment, and their correct production is crucial for different stages of fertilization [56]. However, the overproduction and imbalance of ROS causes oxidative stress, which is highly damaging to proteins, lipids, and DNA [55,56]. When the sperm cell fails to counteract effectively such effects, damaged DNA could promote HPV DNA integration [54,56]. Oxidative stress can be favored by genital tract infections, which induce an increase in seminal fluid leukocytes, capable of producing ROS [57]. Recently, it has been hypothesized to monitor the effect of various antioxidants based on SDF, but further studies are needed to propose it as a potential solution [56].

## 4. Could HPV-Induced Anti-Sperm Antibodies (ASAs) Influence Male Fertility?

The role of anti-sperm antibodies (ASAs) in infertility was highlighted approximately 70 years ago by Wilson [58] and subsequently by Rumke et al. [59], who described the agglutination of serum sperm in infertile men. More recently [60] ASAs have been linked to aberrant semen parameters, such as reduced sperm count and concentration, as well as alterations in motility and membrane integrity. 

Other explanations for ASA-related infertility include induction of sperm agglutination, impairment of sperm transit through the cervical mucus, sperm damage caused by complement activation in the female genital tract, and interference with gametes interaction [61]. 

However, the role of ASAs in fertility remains questionable, with Vazquez-Levin et al. [62] claiming that their presence reduces the likelihood of spontaneous pregnancy and Leushuis et al. [63] arguing that ASAs do not contribute to the reduction of spontaneous conception in sub-fertile couples. Despite these considerations, the WHO Laboratory Manual includes some screening tests for ASAs with a cutoff to assess impaired in vivo fertilization of 50% or more of motile spermatozoa with antibody bound [23,64]. ASAs in semen can be classified among the immunoglobulin classes IgA and IgG; IgA positivity was associated with IgG positivity in 95% of cases, although the latter was less clinically important than the former [23]. 

Risk factors for ASAs development consist of varicocele, prostatitis, orchitis, surgical trauma, unsafe oral or anal sexual practices, breakdown of the blood–testis barrier, testicular cancer, and a variety of microbial infections [65]. Nonetheless, the ASAs’ association with infectious diseases is under debate, with Marconi et al. [66] reporting no clear role in male infertility. Instead, Garolla and colleagues [21] found that more than 40% of infertile patients with HPV infection had ASAs bound to the sperm surface, whereas the infertile patients who were not infected had a lower percentage of antibodies. In addition, the authors suggested that the presence of HPV-DNA on the sperm surface may induce an antigenic response with ASAs formation [21]. 

A few years later, Foresta et al. [29] reported a strong correlation between asthenozoospermia, ASAs, and HPV sperm infection, regardless of the genotype, and further described in a subsequent study by Garolla et al. [67]. Indeed, in several men with idiopathic asthenozoospermia, the presence of HPV DNA in semen appears to be the only risk factor detectable [29] (Table 1). Similarly, Piroozmand et al. [31] showed that 15.2% of infertile men produced ASAs and 17.4% were positive for HPV infection, suggesting that young couples might be tested for HPV and ASAs together with other infertility parameters. All these reports indicate that ASAs have a negative impact on sperm parameters and semen quality, and only the use of some fertility treatments appears to be effective in overcoming the problem of ASAs in semen [68]. 

A better understanding of the mechanisms and role of ASAs and the Implementation of scientific research on this topic would be of great help not only to men with HPV infection but also to their partners, especially if the couple has fertility problems.

## 5. Male Infertility: Differences between hrHPV and lrHPV Infections

Despite the rise of studies investigating the possible role of HPV in male infertility, the distinction between lrHPV and hrHPV infections and their possibly different impacts on sperm parameters have been compared only in a few studies [69]. 

In particular, it was demonstrated that mostly hrHPV genotypes impair sperm parameters such as sperm progressive motility and SDF [70]. In this regard, Piroozmand et al. [31] reported that hrHPV infection significantly reduced sperm count in infertile men, while Wang et al. [40] showed in their metanalysis that infection with hrHPV can lead to a significant reduction in sperm count compared to infection with lrHPV (Table 1). Meanwhile, in a recent study by Capra et al., a higher DFI rate was observed in hrHPV-infected samples compared to lrHPV-infected ones [8].

On the contrary, Rintala et al. [34] found that neither hrHPVs nor lrHPVs have an impact on sperm quality. 

More specifically, it is thought that HPV 16 and 31 E6/E7 proteins may cause sperm genomic disruption or increase apoptotic events, leading to failure of embryo development and worsening potential fertility [32]. 

Men infected with hrHPV genotypes seem to clear the seminal infection over a longer period, resulting in a potential long-term transmission of the virus. In particular, semen infected with HPV 16 (hrHPV) has been shown to act as a vehicle for transporting HPV from the male reproductive tract to the feminine cervix/uterus [24]. 

Concerning lrHPV infections, there does not seem to be any alteration in the canonical sperm parameters such as morphology and motility, but evidence in the literature is scarce. In particular, the absence of E6 and E7 proteins in sperm infected with lrHPV genotypes may explain the lack of effect on sperm parameters. Even if a significantly higher prevalence of oligozoospermia was found in men infected with an lrHPV genotype, this reduction in sperm count does not appear to be a side effect of the infection [71]. 

Moreover, it is thought that the slow replication rate of lrHPV genotypes may cause damage during the late stages of embryo and placental development, reducing the possibility of early abortion, which is more common in infections with hrHPV genotypes, characterized there by more rapid replication [28]. 

## 6. The Potential Role of HPV Infection in Assisted Reproductive Technologies (ART)

Infertility evaluation is recommended if a couple is unable to conceive after twelve or more months of unprotected intercourse with regular frequency [2]. In order to treat infertility effectively, it is necessary to use methods that target the cause of infertility and are most likely to result in pregnancy and birth [72]. Current treatment strategies include pharmacological therapy, surgical therapy, and assisted reproductive technology (ART) [72], which includes intrauterine insemination (IUI) with in vivo fertilization, IVF, and intracytoplasmic sperm injection (ICSI). Specifically, the latter is recommended in cases of semen quality impairment [73]. As mentioned above, HPV infection could have a negative effect on semen quality and motility, and it was also shown that the presence of the virus on the sperm surface could reduce its ability to penetrate the oocyte [18]. Therefore, in an attempt to remove viral DNA, various sperm-washing procedures have been proposed, such as the swim-up method, microfluidic sperm sorting, magnetically activated cell sorting, and density gradient centrifugation [10]. It is noteworthy that one study found that the direct swim-up method was ineffective, while the swim-up technique using heparinase III was the most successful method [74]. 

Another theory regarding viral DNA transmission from spermatozoa to the early embryo has been demonstrated by Mastora et al. [30]. In their study, it is shown how the injection of HPV-positive sperm into the oocyte may negatively impact the outcome of ART since it alters fertilization, implantation, and development of the embryo (Table 1). In particular, it is believed that HPV can interfere with the acrosome reaction, the interaction between sperm and oocyte, and their fusion during ART [75,76]. In fact, in infertile couples, there was a highly statistically significant correlation between the rate of pregnancy loss and the male partner’s positive HPV-DNA test [77]. Perino et al. [78] showed a lower pregnancy rate and higher abortion rate among HPV-positive couples undergoing ART cycles compared to HPV-negative couples. Similarly, Garolla et al. [79] compared the pregnancy rate of two groups of couples undergoing ICSI, divided according to HPV sperm positivity. They showed that 40.8% of the HPV-negative group couples achieved pregnancy, while only 18.2% of the HPV-positive group did. The percentage of blastocyst formation was also lower in the positive group (27.3%) than in the negative group (54.1%), suggesting a negative effect of the virus during early embryonic development. Thus, it is often suggested that young infertile couples with HPV sperm infection delay fertility treatment for 6 months to facilitate clearance [71]. In addition, the detection of HPV in sperm from cryovials commonly used in ART such as ICSI suggests that HPV-DNA can survive under freezing conditions, thus compromising treatment success [75,80]. 

In light of the above, it could be proposed that HPV-positive semen samples should not be used for assisted reproduction or sperm banking and that HPV testing should be performed on sperm donors, couples undergoing ART, and in the general population exhibiting infertility problems [10,15].

## 7. HPV Vaccination—Improvement or Worsening of Male Fertility?

Prophylactic vaccination is the most effective weapon for both men and women in the fight against HPV infection and associated diseases [22]. 

Currently, three vaccines are globally available against HPV—2-, 4- and 9-valent—approved by the Food and Drug Administration (FDA) and subsequently by the European Medicines Agency (EMA) [81,82]. The bivalent, Cervarix (GlaxoSmithKline, Brentford, UK), protects against HPV16 and HPV18 infection, giving potential protection against 70% of cervical cancers but not genital condylomas; the quadrivalent Gardasil (Merck & Co. Inc., Rahway, NJ, USA) potentially protects against both HPV-related manifestations, as it covers HPV types 6, 11, 16 and 18 [83,84]. Lastly, the nonavalent Gardasil9 (Merck & Co. Inc.) has extended the range of protection, as it includes HPV types 6, 11, 16, 18, 31, 33, 45, 52, and 58 [85]. Since it has been shown that male fertility is susceptible to HPV infection, the idea of widespread vaccination not only for women but also for men could improve reproductive health [86].

A recent study showed that some HPV genotypes, such as HPV45, 52, 18, 59, and 16, which are common in male infections, are strongly associated with infertility [76]. Another paper described the prevalence of HPV infection in couples undergoing IVF and showed that HPV52 was often detected in semen samples [87]. Thus, the above genotypes may play an important role in male infertility and, as they are included in the 9-valent vaccine, except the HPV59, vaccination may be a relevant weapon against male infertility, as supported by the current literature [88]. 

For instance, Foresta et al. [89] showed a shorter mean clearance time after vaccination in patients with semen HPV infection. Furthermore, an increase in pregnancies and delivery rates of healthy newborns and a decrease in the miscarriage rate have been observed in association with HPV vaccination of infected men. Notably, an improvement in sperm motility, a reduction in the detection of HPV in sperm and lower ASAs levels have been observed in vaccinated patients, as the absence of HPV on sperm is the parameter that best predicts the delivery rate and pregnancy outcome [22]. In several countries, the effectiveness of HPV vaccination remains debated, and there is a lot of misinformation. 

Initially, vaccination campaigns were targeted at women, aiming to reduce cervical cancer incidence [90]. This has led to significant disparities in vaccination rates between men and women [89]. However, many high-income countries, such as Sweden, the UK, and Canada, are enforcing gender-neutral vaccination programs, and the hope is that by 2030, all European countries will have a gender-neutral approach [90]. Various programs should be implemented to encourage people to get vaccinated [82]; in particular, promoting HPV vaccination among men would be a chance to improve reproductive health in infertile couples whose male partner has a semen infection and to reduce the risk of cancer [91]. Vaccination could also be a key element in facilitating viral infection resolution and revolutionizing the treatment of infertile patients by making them eligible for ART, as it stimulates the development of a humoral immune response that could prevent viral infection [89]. In summary, HPV vaccination could prove to be an effective weapon in the recovery of spermatozoa from HPV infection and therefore in improving the outcomes of natural and assisted reproduction [22].

## 8. Does Unsuccessful Fertilization Depend on Both Partners Being Positive? Or Just One of Them?

It is believed that the reproductive health of both partners, influenced also by STIs, can jeopardize the success of fertilization. In the previous sections, we talked about how HPV-infected sperm can impair fertilization both by affecting semen properties such as motility, pH, and SDF and by transferring viral DNA from infected sperm to the developing embryo [10]. 

Women inseminated with infected sperm seem to have a higher risk of miscarriage, probably because the virus is transmitted to the trophoblast cells, reducing the blastocyst’s potential for implantation [75]. These considerations have already been taken into account in a recent study [79] showing that couples with HPV-infected semen who underwent ART had lower rates of spontaneous and assisted pregnancy and higher rates of abortion. However, a recent meta-analysis [92] has revealed the possible mechanisms by which HPV, mostly hrHPV genotypes, can alter female fertility. 

Infertile women showed higher rates of cervical alterations compared to women with normal fertility status, and in the case of hrHPVs positivity, a higher incidence of cervical intraepithelial neoplasia (CIN) lesions was registered [92]. On the other hand, HPV has also been detected in placental tissue and has been linked to placental insufficiency, premature membrane rupture, preterm birth, and adverse perinatal outcomes. The latter is driven by vertical transmission during vaginal or cesarean delivery with a prolonged rupture of membranes [28]. 

In contrast, Ferrarez et al. [93] described the role of HPV infection in spontaneous miscarriage as not clear, claiming that the presence of HPV is not associated with adverse pregnancy outcomes, and identifying the culprits in older age and possible previous miscarriage. However, although there is little evidence in the literature dealing with the fertility of both positive partners, given what HPV infection means for fertility in both men and women, we can hypothesize that the positivity of one of the two partners will influence the success of fertilization and that therefore, simultaneous positivity in the couple could have a greater effect. 

## 9. Clinical Recommendations and Management

The data described so far highlight the need for counselling strategies for infected couples, especially if they are trying to conceive. Firstly, after diagnosis of HPV positivity in both partners, it is possible to proceed with vaccination and HPV testing every six months for two years in addition to applying some behavioral recommendations, such as genital and hand hygiene, use of personal underwear and towels, avoidance of oral and anal sex [69,94]. It has been described how couples who followed these instructions showed faster viral clearance and shorter persistence of infection compared to couples who did not [69].

As previously reported, men undergoing ART and sperm bank donors should be tested for HPV [10,15]. Particularly, in couples with fertility problems, an HPV screening DNA detection and genotyping from penile swabs and especially from seminal fluid is suggested in the male partner in case of unexplained infertility (not associated with factors of either partner), history of HPV positivity or related manifestations, presence of ASA, and asthenozoospermia [21,69,80].

When HPV positivity is detected, the path to follow will be different from that of the general population. In particular, a fluorescence in situ hybridization (FISH) analysis is recommended to check for the presence of HPV DNA on the sperm surface, while colposcope examination is used to help exclude subclinical genital lesions [29,69,95]. If the result is negative, the couple can try to conceive naturally or through ART [29,69]. If the result is positive, couples counseling is recommended to discuss how to proceed depending on the couple’s age. For older couples, when it is not possible to delay fertility treatment, it is advisable to proceed with ART, particularly after sperm treatment with heparinase III or hyaluronidase. On the contrary, for younger couples, it is recommended to proceed with adjuvant vaccination, a six-month HPV DNA testing follow-up, and FISH analysis to assess viral clearance. If the couple’s results are negative, they may recur to natural fertilization or ART after this test. On the other hand, if positivity persists, it is advisable to repeat the six-month follow-up or to resort to the same strategy of the older couples [10,29,69]. The Flow chart for the management of patients with fertility problems is summarized in Figure 2.

Given the conflicting nature of the literature, further studies are needed to improve the quality of evidence, so to demonstrate the utility of the described strategy and promoting the adoption of interventions necessary to control and limit HPV infection’s impact on reproductive health.

## 10. Future Research Directions and Perspectives

The evidence reported in this review highlights the many roles that HPV infection plays in human fertility and shows the numerous ways in which the virus can affect reproduction. In particular, its impact on the success of spontaneous and assisted reproduction could be related to the alteration of sperm parameters, induction of DNA damage, and genomic instability. The HPV positivity of one or both partners and the different effects of infection with hrHPV or lrHPV genotypes on the fertilization process make prophylactic anti-HPV vaccines an effective weapon against virus-related damage. Therefore, to reduce male-to-female HPV transmission and miscarriage rates, the implementation of gender-neutral vaccination programs may reduce HPV-associated malignancies and reproductive issues [10,86]. In addition, to improve human reproductive health, HPV screening should be recommended for both members of infertile couples [92]. 

Lastly, the principal issue in understanding the impact of HPV on male fertility lies on the scarce knowledge regarding the pathophysiology of certain infertility conditions due to a lack of clinical data [10]. Therefore, future trends in scientific research must focus on widening the number of case studies, trials of new treatments, long-term studies about HPV clearance and its association with fertility improvement, and clinical trials comparing pregnancy rates using different techniques of sperm preparation [10,29,96].

## Figures and Tables

**Figure 1 ijms-24-17562-f001:**
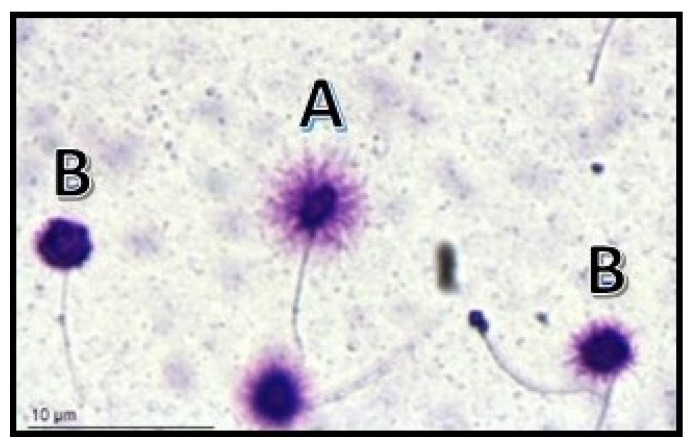
Visualization of sperm DNA fragmentation using sperm chromatin dispersion test (SCD). (A) Normal sperms. (B) Sperms with fragmented DNA.

**Figure 2 ijms-24-17562-f002:**
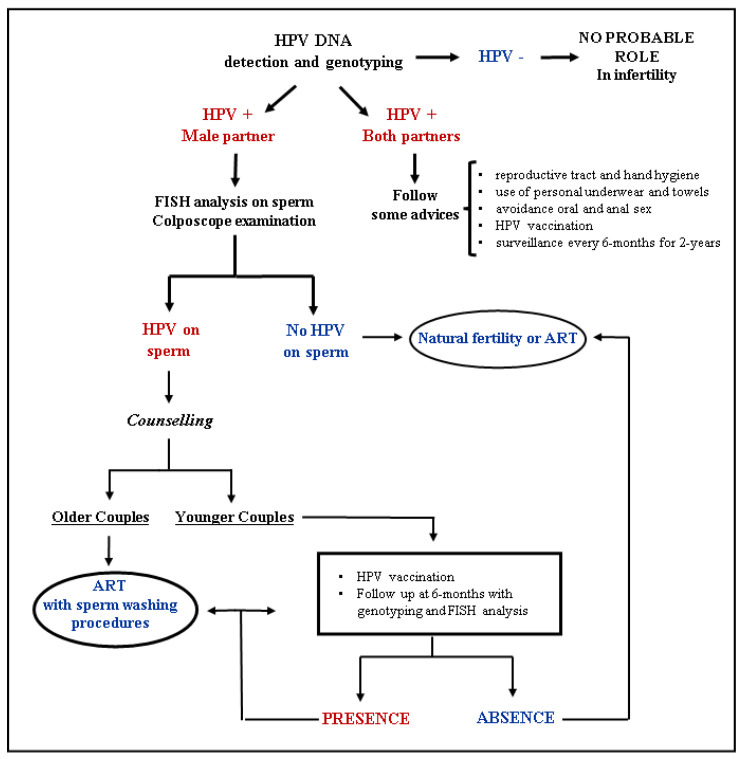
Flow chart for the management of patients with fertility problems. (Adapted with permission from [69], Copyright 2021, Muscianisi, De Toni, Giorato, Carosso, Foresta and Garolla, *Frontiers in Endocrinology*). HPV, Human Papillomavirus; FISH, Fluorescent in situ hybridization; ART, Assisted reproductive techniques.

**Table 1 ijms-24-17562-t001:** Summary of the HPV-related effects on sperm.

Type of Infection	Effect	Reference
HPV positivity	Increased risk of DFI > 30%, asthenozoospermia, ASAs production, and negative ART outcome (alteration in fertilization, implantation, and development of the embryo).	[28,29,30]
hrHPV genotype	Reduced sperm count and motility alterations.	[27,31]
Multiple HPV infections	Hypospermia, abnormal viscosity, and increased seminal pH.	[24]
HPV16 or HPV31	Sperm genomic DNA breaks and increased apoptotic events.	[32]
HPV16 and HPV18	Exonic modification of p53 gene.	[33]

## Data Availability

Not applicable.

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
