# Peer review of "Human Papillomavirus and Male Infertility: What Do We Know?"

_ijms, 2023, doi:10.3390/ijms242417562_

Round 1
Reviewer 1 Report
Comments and Suggestions for Authors
The manuscript titled "Human Papillomavirus and Male Infertility: What Do We Know?" by Arianna Sucato et al. provides a comprehensive review of the existing research on the potential adverse effects of Human Papillomavirus (HPV) infection on male reproductive health. The paper is well-structured, covering various aspects of HPV's impact on male fertility, including its effect on sperm parameters, DNA integrity, and the potential role of HPV-induced anti-sperm antibodies (ASAs) in male fertility.
Here are some suggestions and comments for further improvement:
1. Clarity and Detail in the Introduction: The introduction could benefit from a clearer delineation of the specific aims of the review. While the background information is comprehensive, explicitly stating the main objectives or questions the review seeks to address would enhance its focus.
2. Updated and Comprehensive Literature: Ensure that the literature review is up-to-date and includes the most recent studies in this field. Given the rapidly evolving nature of HPV research, incorporating the latest findings would strengthen the review's relevance and comprehensiveness.
3. Analysis of Conflicting Evidence: The manuscript touches upon conflicting findings in the literature regarding HPV's impact on sperm quality and fertility. A deeper analysis of these discrepancies could be beneficial. Discussing potential reasons for these differences, such as study design, sample size, or methodology, would provide a more nuanced understanding of the topic.
4. Discussion on HPV Vaccination: The review discusses HPV vaccination as a potential strategy to mitigate fertility problems associated with HPV infection. Expanding this section to include more details on current vaccination strategies, their effectiveness in males, and implications for male fertility would be informative. Additionally, discussing any gaps in current vaccination policies that might affect male reproductive health would be valuable.
5. Implications for Clinical Practice: The review could further explore the implications of HPV infection on male infertility in clinical practice. This includes recommendations for screening and management of HPV in infertile males, and how these practices might differ from the general population.
6. Future Research Directions: Concluding with a section on future research directions would be beneficial. This could include identifying gaps in the current understanding of HPV's impact on male fertility, suggesting potential areas for future studies, and discussing the need for longitudinal studies to better understand the long-term effects of HPV on male reproductive health.
Overall, the manuscript is informative and contributes significantly to the understanding of HPV's impact on male fertility. Enhancing certain sections as suggested would make it a more comprehensive resource for readers interested in this field.
Comments on the Quality of English LanguageThe manuscript demonstrates a good level of English language proficiency. The vocabulary is appropriate for a scientific review, and the sentence structure is generally clear and coherent. However, there are occasional instances where the language can be refined for better clarity and readability.
Author Response
First of all, we would like to thank the reviewer for his/her comments and suggestions, that we hope to have totally addressed. You will find all the changes highlighted in the manuscript and a point-by-point answer to each comment below. We have tried to improve the quality of the manuscript, and we are hopeful that this improved version will be suitable for publication.
The manuscript titled "Human Papillomavirus and Male Infertility: What Do We Know?" by Arianna Sucato et al. provides a comprehensive review of the existing research on the potential adverse effects of Human Papillomavirus (HPV) infection on male reproductive health. The paper is well-structured, covering various aspects of HPV's impact on male fertility, including its effect on sperm parameters, DNA integrity, and the potential role of HPV-induced anti-sperm antibodies (ASAs) in male fertility.
Here are some suggestions and comments for further improvement:
- Clarity and Detail in the Introduction: The introduction could benefit from a clearer delineation of the specific aims of the review. While the background information is comprehensive, explicitly stating the main objectives or questions the review seeks to address would enhance its focus.
Thank you for your suggestion, we have included this on page 2, lines 88-93.
- Updated and Comprehensive Literature: Ensure that the literature review is up-to-date and includes the most recent studies in this field. Given the rapidly evolving nature of HPV research, incorporating the latest findings would strengthen the review's relevance and comprehensiveness.
Thank you for your comment. The reference list has been updated.
- Analysis of Conflicting Evidence: The manuscript touches upon conflicting findings in the literature regarding HPV's impact on sperm quality and fertility. A deeper analysis of these discrepancies could be beneficial. Discussing potential reasons for these differences, such as study design, sample size, or methodology, would provide a more nuanced understanding of the topic.
Thank you for your suggestion, we have included this on page 3, lines 113-121.
- Discussion on HPV Vaccination: The review discusses HPV vaccination as a potential strategy to mitigate fertility problems associated with HPV infection. Expanding this section to include more details on current vaccination strategies, their effectiveness in males, and implications for male fertility would be informative. Additionally, discussing any gaps in current vaccination policies that might affect male reproductive health would be valuable.
Thank you for your observation, we have included this on page 8, lines 349-360.
- Implications for Clinical Practice: The review could further explore the implications of HPV infection on male infertility in clinical practice. This includes recommendations for screening and management of HPV in infertile males, and how these practices might differ from the general population.
Thank you for your suggestion. We have included a new section and an image about this topic on pages 9 and 10.
- Future Research Directions: Concluding with a section on future research directions would be beneficial. This could include identifying gaps in the current understanding of HPV's impact on male fertility, suggesting potential areas for future studies, and discussing the need for longitudinal studies to better understand the long-term effects of HPV on male reproductive health.
Thank you for your comment. We have included this topic in the last section on page 11, lines 448-453.
Overall, the manuscript is informative and contributes significantly to the understanding of HPV's impact on male fertility. Enhancing certain sections as suggested would make it a more comprehensive resource for readers interested in this field.
Reviewer 2 Report
Comments and Suggestions for Authors
In this interesting paper, the authors reviewed the increasing attention to understanding the causes of infertility, recognized as a growing health problem affecting large numbers of couples worldwide. Indeed, male infertility is a contributing factor in approximately 30-40% of cases, and one of its etiological causes is sexually transmitted infections. Among sexually transmitted pathogens, HPV can contribute in various ways to the failure of spontaneous and assisted reproduction, acting in the different phases of conception, especially in the early ones. In particular, HPV infection can affect sperm DNA integrity, sperm motility, count, viability, and morphology, and can induce the production of anti-sperm antibodies. The authors aimed to provide an overview of existing research on the potential adverse effects of HPV infection on male reproductive health. The authors also described how limiting the spread of the infection, particularly with gender-neutral vaccination, could be a possible therapeutic tool to counteract male and female fertility problems.
The paper is generally well-written with a few English inaccuracies. However, in the present form, a reader may raise a few concerns.
- The authors should describe the novelty of their work since only in PubMed, I found a dozen of similar reviews.
- I do believe that this paper is a “narrative review”. This is not stated. Please explain.
- There is no description of the reference selection methods.
- Compared to similar reviews dealing with male infertility and HPV, this paper is characterized by the absence of an iconographic section(s). Why? This is quite weird. The inclusion of Figure(s) could potentiate both the aim and message of the authors.
- The reference list is short and could be updated by including further papers.
- The role of elevation in oxidative stress in male infertility and HPV sounds poor and could be updated with further info.
- A similar criticism may be raised for Asthenozoospermia and HPV.
Minor editing of the English language required
Author Response
First of all, we would like to thank the reviewer for his/her comments and suggestions, that we hope to have totally addressed. You will find all the changes highlighted in the manuscript and a point-by-point answer to each comment below. We have tried to improve the quality of the manuscript, and we are hopeful that this improved version will be suitable for publication.
In this interesting paper, the authors reviewed the increasing attention to understanding the causes of infertility, recognized as a growing health problem affecting large numbers of couples worldwide. Indeed, male infertility is a contributing factor in approximately 30-40% of cases, and one of its etiological causes is sexually transmitted infections. Among sexually transmitted pathogens, HPV can contribute in various ways to the failure of spontaneous and assisted reproduction, acting in the different phases of conception, especially in the early ones. In particular, HPV infection can affect sperm DNA integrity, sperm motility, count, viability, and morphology, and can induce the production of anti-sperm antibodies. The authors aimed to provide an overview of existing research on the potential adverse effects of HPV infection on male reproductive health. The authors also described how limiting the spread of the infection, particularly with gender-neutral vaccination, could be a possible therapeutic tool to counteract male and female fertility problems.
The paper is generally well-written with a few English inaccuracies. However, in the present form, a reader may raise a few concerns.
- The authors should describe the novelty of their work since only in PubMed, I found a dozen of similar reviews.
Thank you for your comment. This information has been added on page 2, lines 88-93.
- I do believe that this paper is a “narrative review”. This is not stated. Please explain.
Thank you for your suggestion, we have included this on page 1, line 24, and page 2, lines 88-93.
- There is no description of the reference selection methods.
Thank you for your observation, but as you suggested in the previous point, we are only proposing a "narrative review", not a "systematic review of the literature". Therefore, we felt it was unnecessary to describe the selection methods.
- Compared to similar reviews dealing with male infertility and HPV, this paper is characterized by the absence of an iconographic section(s). Why? This is quite weird. The inclusion of Figure(s) could potentiate both the aim and message of the authors.
Thank you for your suggestion. Two images have been added on pages 5 and 10.
- The reference list is short and could be updated by including further papers.
Thank you for your comment. The reference list has been updated.
- The role of elevation in oxidative stress in male infertility and HPV sounds poor and could be updated with further info.
Thank you for your observation, we have included this on page 4, lines 187-195.
- A similar criticism may be raised for Asthenozoospermia and HPV.
Thank you for your suggestion, we have included this on page 5, lines 235-238.
Round 2
Reviewer 1 Report
Comments and Suggestions for Authors
Accept in present form
Author Response
We thank the reviewer for his advice and for considering this improved version suitable for publication.
Reviewer 2 Report
Comments and Suggestions for Authors
- The authors should describe the novelty of their work since only in PubMed, I found a dozen of similar reviews.
Thank you for your comment. This information has been added on page 2, lines 88-93.
Intriguing answer. Ok
- I do believe that this paper is a “narrative review”. This is not stated. Please explain.
Thank you for your suggestion, we have included this on page 1, line 24, and page 2, lines 88-93.
Answer Ok
- There is no description of the reference selection methods.
Thank you for your observation, but as you suggested in the previous point, we are only proposing a "narrative review", not a "systematic review of the literature". Therefore, we felt it was unnecessary to describe the selection methods.
Does this mean that you included papers from the International Journal of Mickey Mouse? Please include a few lines showing the origin and reason of paper selection.
- Compared to similar reviews dealing with male infertility and HPV, this paper is characterized by the absence of an iconographic section(s). Why? This is quite weird. The inclusion of Figure(s) could potentiate both the aim and message of the authors.
Thank you for your suggestion. Two images have been added on pages 5 and 10.
Partially Ok. Figure 2 is quite poor in the pixel quality and should be improved.
- The reference list is short and could be updated by including further papers.
Thank you for your comment. The reference list has been updated.
Answer Ok
- The role of elevation in oxidative stress in male infertility and HPV sounds poor and could be updated with further info.
Thank you for your observation, we have included this on page 4, lines 187-195.
Answer Ok. Thank you
- A similar criticism may be raised for Asthenozoospermia and HPV.
Thank you for your suggestion, we have included this on page 5, lines 235-238.
Answer Ok
Author Response
Does this mean that you included papers from the International Journal of Mickey Mouse? Please include a few lines showing the origin and reason of paper selection.
Done. Page 2, lines 90-94.
Partially Ok. Figure 2 is quite poor in the pixel quality and should be improved.
We have tried to improve the quality of the image by saving it as a TIFF file and changing the font and color. We hope that this improved version will be suitable.
Round 3
Reviewer 2 Report
Comments and Suggestions for Authors
In the file I downloaded Figure 2 is still ugly and MUST be improved. Please ask for the support of a professional in this field